# CONFLATOR: Incorporating Switching Point based Rotatory Positional Encodings for Code-Mixed Language Modeling

**Mohsin Ali[1]    Sai Teja Kandukuri[2]    Neeharika Gupta[3]    Parth Patwa[4]**
**Anubhab Chatterjee[3]    Vinija Jain[5]    Aman Chadha[5,6*]    Amitava Das[7]**
[1]San Diego State University, USA    [2]San Jose State University, USA
[3]Wipro AI Labs, India    [4]University of California Los Angeles, USA
[5]Stanford University, USA    [6]Amazon AI, USA    [7]University of South Carolina, USA
[1]mmohammed5956@sdsu.edu    [2]saiteja.kandukuri@sjsu.edu
[7]amitava@mailbox.sc.edu

## Abstract

The mixing of two or more languages is called Code-Mixing (CM). CM is a social norm in multilingual societies. Neural Language Models (NLMs) like transformers have been effective on many NLP tasks. However, NLM for CM is an under-explored area. Though transformers are capable and powerful, they cannot always encode positional information since they are non-recurrent. Therefore, to enrich word information and incorporate positional information, positional encoding is defined. We hypothesize that Switching Points (SPs), i.e., junctions in the text where the language switches (L1 → L2 or L2 → L1), pose a challenge for CM Language Models (LMs), and hence give special emphasis to SPs in the modeling process. We experiment with several positional encoding mechanisms and show that rotatory positional encodings along with switching point information yield the best results.

We introduce CONFLATOR: a neural language modeling approach for code-mixed languages. CONFLATOR tries to learn to emphasize switching points using smarter positional encoding, both at unigram and bigram levels. CONFLATOR outperforms the state-of-the-art on two tasks based on code-mixed Hindi and English (Hinglish): (i) sentiment analysis and (ii) machine translation.

## 1   Code-Mixing: Juxtaposition of two Languages

Code-mixing is defined as the alternation of two or more languages during articulation. Recently, code-mixing has gained a lot of attention in the area of NLP due to the prevalence of language mixing in multilingual societies such as India, Europe, US, South Africa, Mexico, etc. In such societies, code-mixing is fairly commonplace, especially in informal conversations, where the native language

is often romanized and code-mixed with an auxiliary language. This effect occasionally manifests in posts originating from the aforementioned sources on social media platforms such as Twitter, Facebook, etc. An example of Hindi and English code-mixing is shown in the following phrase where an English word, *dance*, is mixed with Hindi romanized words: *Gaaye, aur, kare*.

$Gaaye_{HI}\ aur_{HI}\ dance_{EN}\ kare_{HI}$
**English translation:** sing and dance

With the proliferation of code-mixing on the internet, it is important to study language processing and language modeling for code-mixed languages. While language modeling using neural networks has come a long way, replacing n-gram language models with distributed neural representations (Bengio et al., 2003) to recent large transformer-based pre-trained language models (LMs) such as GPT-x (Radford et al., 2019), BERT (Devlin et al., 2018a) etc., code-mixed language modeling using state-of-the-art (SoTA) Transformer-based models is still under-explored.

The biggest hindrance in the adoption of SoTA Transformer-based LMs for code-mixing can be attributed to data scarcity. While Transformer-based (Vaswani et al., 2017b) architectures such as BERT and GPT have set new benchmarks in the domain of language modeling, they are infamous for their low sample efficiency. In other words, the voracious data appetite of Transformers and the lack of substantial code-mixed datasets in the community is the primary reason for the technological hindrances in the area of code-mixed language modeling compared to vanilla language modeling.

To corroborate the aforementioned arguments, we experiment with Transformer-based models such as GPT-2 and BERT for code-mixing. We empirically observe that these models perform poorly on tasks involving code-mixed data. Our hypothesis is as follows: Since information related to switch-

---

*Work does not relate to position at Amazon.

ing point is a major component in the context of code-mixed content, it should thus be incorporated in downstream processing. Switching points are a bottleneck for a model's processing of code-mixed data and the reason for poor performance using SoTA neural language models (Chatterjere et al., 2020). Switching points play a crucial factor when dealing with CM data. In the next few sections, we discuss various positional encoding approaches, switching points, and our approaches for language modeling on code-mixed data. Our key contributions are:

- We propose *CONFLATOR*, an LM system that incorporates switching point related positional information.

- Our system improves the performance of existing models and achieves a new SoTA on two tasks.

- We investigate, experiment with, and introduce various switching point based positional encoding techniques.

- We introduce a novel Switching Point based Rotary matrix for Rotary Positional Encoding (RoPE).

- We curate a new dataset of code-mixed tweets.

## 2 Related Work

It is important to study code-mixing as it is a part of most multilingual societies and prevalent in social media. It is more complex to process code-mixed text than monolingual text for NLP tasks (Verma, 1976). Similar line of work was followed by Bokamba (1988) and Singh (1985) on the complexities of multi-languages on the basis of syntactics and grammar. The difficulties of processing code-mixed languages on social media is further exacerbated by unusual spellings, many unique ways of writing the same word, unnecessary capitalization etc (Das and Gambäck, 2014; Laddha et al., 2020).

With the growing popularity on social media, Various tasks like sentiment analysis (Patwa et al., 2020a; Chakravarthi et al., 2020), translation (Dhar et al., 2018; Srivastava and Singh, 2020), hate-speech detection (Bohra et al., 2018; Banerjee et al., 2020), POS tagging (Vyas et al., 2014), etc. have been performed on code-mixed data. Methods to handle code-mixing for text classification include the use of CNNs (Aroyehun and Gelbukh, 2018; Patwa et al., 2020b), Transformer or BERT like

models (Samghabadi et al., 2020; Tang et al., 2020), ensemble models (Tula et al., 2021; Jhanwar and Das, 2018), focal loss (Tula et al., 2022; Ma et al., 2020) etc.

Vaswani et al. (2017a) proposed transformers for neural language modeling using masked language modeling (MLM) and next sentence prediction, which achieved SoTA performance on many NLP tasks. Devlin et al. (2018b) released mBERT, a model trained on multilingual corpus that includes 104 languages. A cross lingual language model XLM was proposed in Lample and Conneau (2019) which leveraged monolingual and crosslingual corpus for pretraining. Nayak and Joshi (2022) present a bert pretrained on CM data. However, they do not make changes to their language model or technique to handle code-mixed data in particular. Sengupta et al. (2021) propose a Hierarchical transformer based architecture that captures the semantic relationship among words and hierarchically learns the sentence level semantics of code-mixed data. Ali et al. (2022) Were one of the first to incorporate switching point information in positional encoding. They utilize dynamic positional encodings whereas our method, CONFLATOR infuses switching point information in rotatory positional encodings and also uses both unigram and bigram tokens to get the final embedding.

## 3 Data Extraction and Strategies

In this section, we discuss the details of code-mixed data extraction. Our primary aim is to extract naturally distributed code-mixed data.

### 3.1 Qualitative and Quantitative Checkpoints for Hinglish Corpus

The performance of LMs is dependent on the training data size and quality, along with the vocabulary size. Code-mixed language modeling suffers from the following challenges: i) data scarcity, ii) Words from 2 (or more) languages in the same sentence, iii) *Hindi* is written using *English* letters (i.e. transliteration), hence, there is no standardization of spelling - which in effect proliferates word forms (Laddha et al., 2020, 2022), iii) Code-mixing is usually found on social media and netizens often incorporate creativity in their mixing along with wordplay. We consider two fundamental questions to guide our data collection:

1. *The performance on any NLP task depends on the data complexity:*

**Empirical measurement:** Consider two 4-word tweets - i) $T_i : w_{L1}w_{L1}w_{L2}w_{L2}$ and ii) $T_j : w_{L1}w_{L2}w_{L1}w_{L2}$. Both the tweets have 2 words each from the languages $L1$ and $L2$. Thus the mixing ratio of both the tweets $T_i$ and $T_j$ is $(4-2)/4 = 0.50$. However, $T_i$ only contains 1 code alternation point whereas $T_j$ contains 3 switches. It is likely that $T_j$ is harder to process. Hence, we need a metric for the level of mixing between the languages. We use *Code-Mixing-Index* (Gambäck and Das, 2016) (CMI) to measure such complexity. Please refer to section 3.2 for more details on CMI.

2. *How much data is good enough?*

**Empirical measurement:** When two languages blend, it is quite natural that the number of unique word forms would be much higher in a Hinglish corpus in comparison to monolingual English or Hindi corpus. Therefore, we ask an essential question at the very beginning, *how much data is good enough?* We decide to keep collecting data, until the Heaps' curve starts converging so that we cover most of the unique words.

Heaps' law (Gopalan and Hopkins, 2020) states that the number of unique words in a text of $n$ words is approximated by $V(n) = Kn^{\beta}$ where $K$ is a positive constant and $\beta$ lies between 0 and 1, $K$ invariably lies between 10 and 100 and $\beta$ between 0.4 an 0.6. Heaps' law is often considered to be a good estimator to calculate the vocabulary size. To compare, from the figure 1, it can be seen that, for English Wiki, the flattening of the Heaps' law curve, starts at **40K-50K**, whereas for monolingual Hindi, it converges at **80K-90K**, but for *Hinglish* the same behavior starts around **800K** vocabulary and 50M words.

### 3.2 Code-Mixing Index (CMI)

As mentioned previously, we expect the difficulty of language processing tasks to increase as the level of code-mixing increases. To measure the level of code-mixing in our corpus, we use Code-mixing Index (Gambäck and Das, 2016) :

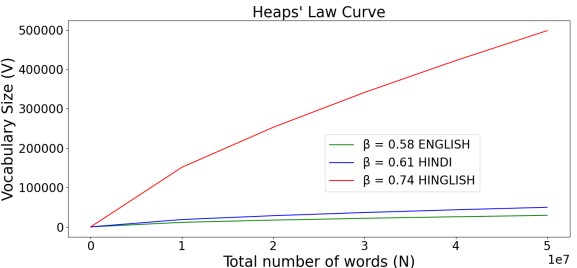

Figure 1: Heaps' plot on 50M word forms in English, Hindi and Hinglish corpora. The $\beta$ values are 0.58, 0.61, 0.74 respectively.

$$
\begin{aligned}
C_u(x) &= w_m f_m(x) + w_p f_p(x) \\
&= w_m \frac{N(x) - \max_{L_i \epsilon L}(tLi)(x)}{N(x)} * 100 + w_p \frac{P(x)}{N(x)} * 100 \\
&= 100 * \frac{w_m((N(x) - \max_{L_i \epsilon L}(tLi)(x)) + w_p P(x)}{N(x)}
\end{aligned}
\tag{1}
$$

Where x denotes utterance, N is the number of token in x belonging to any language $L_i$, $w_m$ and $w_n$ are weights. Please refer to Gambäck and Das (2016) for a detailed explanation of CMI.

### 3.3 Data Acquisition Pipeline

We follow a pipeline similar to (Chatterjere et al., 2020). We collect CM data from Twitter via the Twitter API. We need to use relevant keywords (words unique to Hindi) in our search to get CM tweets. Words with lexical overlap between Hindi and English should not be used for searching. for example, the word *do* is confusing because it means two in Hindi. We start with the *ICON 2017* Hinglish sentiment analysis dataset (Patra et al., 2018), which is annotated with word-level language. From this data, we create two vocabularies $V_{HI}$ and $V_{EN}$, and generate a vocabulary of unique Hindi words $V_{HI-UNIQ} = V_{HI} - I$, where $I = V_{HI} \bigcap V_{EN}$. $V_{HI-UNIQ}$ set is then sorted in descending order, based on the word frequency, and is used as search words on the Twitter API. Once we get the tweets, we use a word-level language identifier (Barman et al., 2014) (having 90%+ accuracy) on the tweets and calculate the CMI of the tweet. Once we get the word-level language labels, we can also know where the switching points are. Tweets with CMI = 0 are discarded. Finally, we are left with 87k tweets. The CMI distribution of our data is given in table 1. This dataset is used to pretrain our models.

**Training and Testing data:** We collect 87K sentences distributed over all CMI ranges, instead of

| CMI | # Tweets | Percentage |
|---|---|---|
| 0-10 | 7,036 | 8.05% |
| 11-20 | 16,481 | 18.9% |
| 21-30 | 22,617 | 25.9% |
| 31-40 | 22,722 | 26.0% |
| 41-50 | 11,404 | 13.1% |
| 50+ | 7,036 | 8.05% |
| Mean CMI: **28** | Total # of tweets: **87,296** | |

Table 1: CMI distribution of the collected data. The total number of extracted tweets is 87K.

collecting equal data across the CMI ranges, so that the resultant languages trained on this corpus would be able to handle real data. We maintain the same distribution over both our training and testing corpora (*4:1 ratio*), for our language models.

## 4 The Bottleneck of Code-mixed Language Modeling: Switching Points

Formally, Switching Points (SPs) are the tokens in text, where the language switches. For code-mixed languages, consisting of a pair of languages, there can be two types of switching points. Suppose the two languages as part of the code-mixed language are *L1* and *L2*, a switching point occurs when the language in the text changes from L1 to L2 or L2 to L1. To explain it better, let us consider the following sample in *Hinglish*:

*gaana*HI *enjoy*EN *kare*HI
**English Translation:** Enjoy the song.

In the above example, when the language switches from *Hindi* to *English* (*gaana*HI *enjoy*EN) a **HI-EN** (HIndi-ENglish) switching point occurs. Similarly, a **EN-HI**(ENglish-HIndi) switching point occurs at - *enjoy*EN *kare*HI.

In the context of modeling code-mixed languages, switching points can be considered as ordinary bigrams, that occur with other monolingual bigrams in a corpus. It is easy to infer that particular SP bigrams will be relatively rare in a given corpus. Hence, such sparse occurrences of switching point bigrams make it difficult for any Language Model to learn their probabilities and context. Since the language changes at the switching point, LMs are likely to find it difficult to process these tokens. In order to counter this challenge, we partition our code-mixed data into **(i)** *switching points*, and **(ii)** *non-switching points*. We then build LMs specifically for switching points and non-switching points,

as discussed in the following sections.

**CONFLATOR Hypothesis:** The CONFLATOR is built on 2 hypotheses. i) Positional information is important for language models, especially when dealing with CM text. ii) Switching points are the bottleneck for code-mixed language models (CMLM). We incorporate positional information of switching points into our CMLM.

## 5 Positional Encoding Techniques

As discussed, SPs are a major bottleneck hence handling them separately is needed. Positional encoding are necessary for language models to learn dependencies between tokens. Positional embedding was first introduced by Vaswani et al. (2017b). The proposed sinusoidal positional encoding is composed of sine and cosine values with position index as inputs. The encoding techniques are further improved by Liu et al. (2020) where a dynamic function is introduced to learn position with gradient flow and Shaw et al. (2018) learned positional representation of relative positions using a learnable parameter. We talk about different positional encoding techniques in detail in the following subsections.

We experiment with several contemporary techniques and find that rotary positional encoding (Su et al., 2021) performs the best.

### 5.1 Sinusoidal Positional Encoding (SPE)

Vaswani et al. (2017b) introduced a pre-defined sinusoidal vector $p_i \in R^d$ which is assigned to each position $i$. This $p_i$ is added to the word embedding $x_i \in R^d$ at position $i$, and $x_i + p_i$ is used as input to the model such that the Transformer can differentiate words coming from different positions and this also assigns each token a position-dependent attention. - equation 2.

$$e_{ij}^{abs} = \frac{1}{\sqrt{d}} \left( (x_i + p_i) W^{Q,1} \right) \left( (x_j + p_j) W^{K,1} \right)^T \quad (2)$$

Where W is the weight matrix, Q is query, K is key, l in the layer.

### 5.2 Dynamic Positional Encoding (DPE)

Instead of using predefined periodical functions like $sin$, Liu et al. (2020), introduced a dynamic function $\Theta(i)$ at every encoder layer. Improving upon sinusoidal PE, Dynamic PE learns $\Theta(i)$ instead of a predefined $p_i$ to bring dynamic behavior to the model. At each utterance, this learnable function $\Theta(i)$ tries to learn the best possible representation for positional information with gradient flow.

$\Theta(i)$ is added to the word embedding $w_i$ as given in equation 3.

$$e_{ij} = \frac{1}{\sqrt{d}} \left( (x_i + \Theta(i)) W^{Q,1} \right) \left( (x_j + \Theta(j)) W^{K,1} \right)^T \quad (3)$$

### 5.3 Relative Positional Encoding (RPE)

In absolute PE, using different $p_i$ for different positions $i$ helps the transformer distinguish words at different positions. However, the absolute PE is not effective in capturing the relative word order. Shaw et al. (2018) introduced a learnable parameter $a_{i-j}^l$ which learns the positional representation of the relative position $i$-$j$ at encoder layer $l$. With the help of this, we can explicitly capture word orders in our model as follows:

$$e_{ij}^{rel} = \frac{1}{\sqrt{d}} \left( (x_i)^l W^{Q,l} \right) \left( (x_i)^l W^{K,l} + a_{i-j}^l \right)^T \quad (4)$$

### 5.4 Switching Point-based Dynamic and Relative Positional Encoding (SPDRPE)

Ali et al. (2022) introduce a novel, switching point based PE. For illustration purposes, consider a code-mixed Hinglish text - *ye_HI gaana_HI enjoy_EN kare_HI*. SP-based indices **(SPI)** set the index to 0 whenever an SP occurs. Indexing would normally be *Index* = (0, 1, 2, 3), but due to switching point incorporation, this gets changed to *SPI* = (0, 1, 0, 0). In addition to this, they use a learning parameter $a_{i-j}^l$, which encodes the relative position $i$-$j$ at the encoder layer $l$. This encoding approach learns representations dynamically based on SPs along with the embedding $a_{i-j}^l$ so that it can also capture relative word orders, as follows:

$$e_{ij} = \frac{1}{\sqrt{d}} \left( (x_i + \Theta(S(l_i)))^l W^{Q,l} \right) \left( (x_i + \Theta(S(l_j)))^l W^{K,l} + a_{i-j}^l \right)^T \quad (5)$$

### 5.5 Rotary Positional Encoding (RoPE)

Analogous to the idea of electromagnetic waves going through a polarizer to preserve their relative amplitude, (Su et al., 2021) came up with the idea of Rotary Positional Encoding (RoPE). The idea is to use rotation matrices on the embedding vectors to generate the positional values. The rotation negates any absolute positional information and only retains information about the relative angles between every pair of word embeddings in a sequence. We know that the dot product between two vectors is a function of the magnitude of individual vectors and the angle between them. Keeping this in mind, the intuition for RoPE is to represent the embeddings as complex numbers and the positions as pure rotations that we apply to them. Mathematically, the formulations for a simple 2-dimensional case are defined as follows:

$$f_Q(x_i, i) = (W_Q x_i) e^{\sqrt{-1} i \theta}$$
$$f_Q(x_j, j) = (W_K x_j) e^{\sqrt{-1} j \theta}$$
$$g(x_i, x_j, i-j) = Re[(W_Q x_i)(W_K x_i)^* e^{\sqrt{-1}(i-j)\theta}]$$
$$(6)$$

where $Re[]$ is the real part of a complex number and $(W_K x_i)^*$ represents the conjugate complex number of $(W_K x_i)$. $\theta \in R$ is a preset non-zero constant. Formulating $f_{(Q,K)}$ as a matrix multiplication, we get:

$$f_Q(x_i, i) = \begin{pmatrix} cosm\theta_1 & -sinm\theta_1 \\ sinm\theta_1 & cosm\theta_1 \end{pmatrix} \begin{pmatrix} W_{Q,K}^{(11)} & W_{Q,K}^{(12)} \\ W_{Q,K}^{(21)} & W_{Q,K}^{(22)} \end{pmatrix} \begin{pmatrix} x_i^{(1)} \\ x_i^{(2)} \end{pmatrix} \quad (7)$$

where $(x_i^{(1)}, x_i^{(2)})$ is $x_i$ expressed in the form of 2D coordinates. In the same way, we can turn function $g$ into matrix form. By rotating the transformed embedding vector by an angle in multiples of its position index, we are able to incorporate relative position information. Due to this characteristic, it is termed as Rotary Position Embedding.

In order to generalize the result in 2D to any $x_i$ in $R_d$ where $d$ is even, they divide the d-dimension space into $\frac{d}{2}$ sub-spaces and combine them in merit of the linearity of inner product, turning the attention formulation:

$$f_{Q,K} = e_{ij}^{rotary} = \frac{1}{\sqrt{d}} \left( RM_{\Theta,i}^d W^{Q,1}(x_i) \right)^T \left( RM_{\Theta,j}^d W^{K,1}(x_j) \right) \quad (8)$$

$$RM = \begin{pmatrix} cosm\theta_1 & -sinm\theta_1 & 0 & 0 & ... & 0 & 0 \\ sinm\theta_1 & cosm\theta_1 & 0 & 0 & ... & 0 & 0 \\ 0 & 0 & cosm\theta_2 & -sinm\theta_2 & ... & 0 & 0 \\ 0 & 0 & sinm\theta_2 & cosm\theta_2 & ... & 0 & 0 \\ . & . & . & . & ... & . & . \\ . & . & . & . & ... & . & . \\ 0 & 0 & 0 & 0 & ... & cosm\theta_{d/2} & -sinm\theta_{d/2} \\ 0 & 0 & 0 & 0 & ... & sinm\theta_{d/2} & cosm\theta_{d/2} \end{pmatrix} \quad (9)$$

where RM is orthogonal and sparse matrix predefined parameters

$$\Theta = \theta_i = 10000^{-2(i-1)/d}, i \in [1, 2, ..., d/2]. \quad (10)$$

In contrast to the additive nature of the position embedding methods used by other works, their approach is multiplicative. Moreover, RoPE naturally incorporates relative position information through rotation matrix product instead of altering terms in the expanded formulation of additive position encoding when applied with self-attention.

## 6 Incorporation of Switching Point Information in CMLM

Positional encodings help the transformer learn dependencies between tokens at different positions of the input sequence. To enhance the positional encodings for code-mixed text, we modify the rotatory positional encoding to incorporate switching point information.

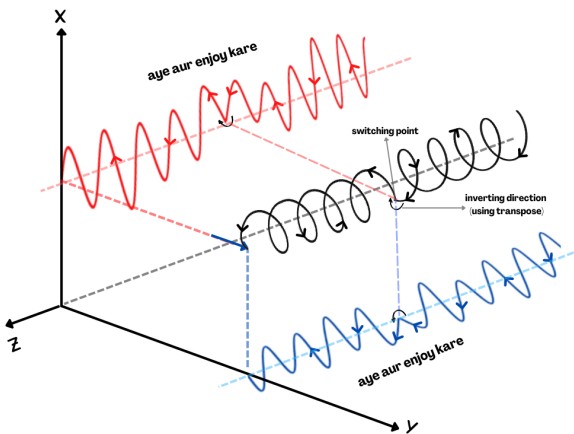

Figure 2: Visual intuition for our rotary approach with switching point incorporation. We consider a linearly polarized electromagnetic wave and show the change in rotation whenever a switching point occurs.

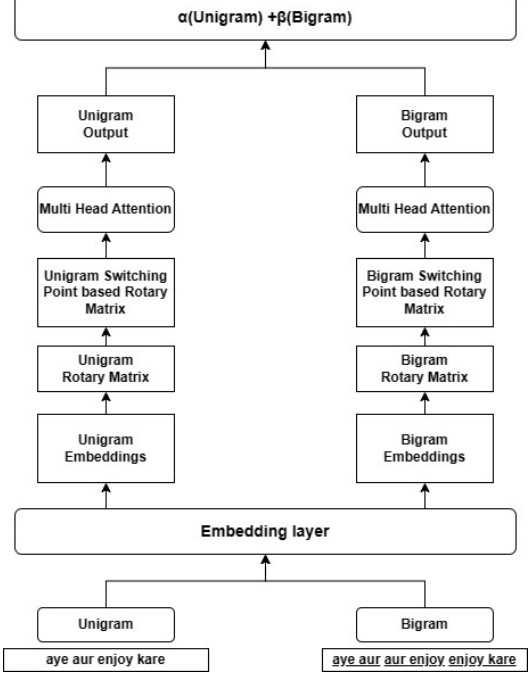

Figure 3: This diagram depicts the higher level understanding of the proposed positional embeddings.

## 6.1 Switching Point-based Rotary Matrix

Switching points are a potential bottleneck for code-mixing language modeling and to address this problem, we incorporate switching point based rotary positional encoding in our architecture. The intuition behind RoPE is electromagnetic waves. The embeddings are represented as complex numbers and the positions are represented as pure rotations that are applied to them. Keeping this in mind, we address the problem of switching points (SP) with the help of angles that participate in RoPE. Whenever we encounter a switching point, we change the

rotation, i.e., we change the direction of these angles. To implement the rotation change, we define a switching point matrix. The switching point matrix helps our model identify and learn the patterns of code mixing in the corpus. Our matrix is defined with 1s and -1s. When there is a language shift (L1 → L2) or (L2 → L1), i.e., when we encounter a switching point, we annotate the column value as -1 and for the successive words in L2, we annotate column values as 1 until another switching point occurs.

$$SPM \in R_{n*n}^d$$
$$\text{if } i == SP:$$
$$SPM_i = -1 \quad (11)$$
$$\text{else:}$$
$$SPM_i = 1$$

The visual intuition of our approach is shown in Figure 2. The switching point matrix (SPM) with 1s and -1s is defined in such a way that it transposes the rotary matrix, intuitively inverting the rotation at every switching point encounter. Therefore, the final matrix, i.e., switching point rotary matrix (SPRM) is a result of element-wise multiplication of the defined switching point matrix (SPM) with rotary matrix (RM):

$$SPRM = SPM \times RM \quad (12)$$

$$e_{ij}^{SPRotary} = \frac{1}{\sqrt{d}} \left( SPRM_{\Theta,i}^d W^{Q,1}(x_i) \right)^T \left( SPRM_{\Theta,j}^d W^{K,1}(x_i) \right) \quad (13)$$

## 6.2 Bigram and Switching Point-based Rotary Positional Encoding (BSPRoPE)

Since the language changes at the SPs, we get two consecutive tokens with different language hence we also incorporate the bigram level information in our model. In this positional encoding method, we get positional information among the bigrams in an utterance. We use the technique of switching point based rotary positional encoding at a word-to-word level and at bigram level as depicted in Figures 3,4 and mathematically expressed as Equation 16

$$e_{ij}^{UniSPRotary} = \frac{1}{\sqrt{d}} \left( SPRM_{\Theta,i}^d W^{Q,1}(x_i) \right)^T \left( SPRM_{\Theta,j}^d W^{K,1}(x_j) \right) \quad (14)$$

$$e_{ij}^{BiSPRotary} = \frac{1}{\sqrt{d}} \left( SPRM_{\Theta,i}^d W^{Q,1}(x_i) \right)^T \left( SPRM_{\Theta,j}^d W^{K,1}(x_j) \right) \quad (15)$$

$$prediction = a * e_{ij}^{UnigramSPRotary} + b * e_{ij}^{BigramSPRotary} \quad (16)$$

where $a$ and $b$ are learnable coefficients. $x_i$ and $x_i$ in equation 14 refer to unigram inputs whereas as in equation 15 they refer to bigram inputs.

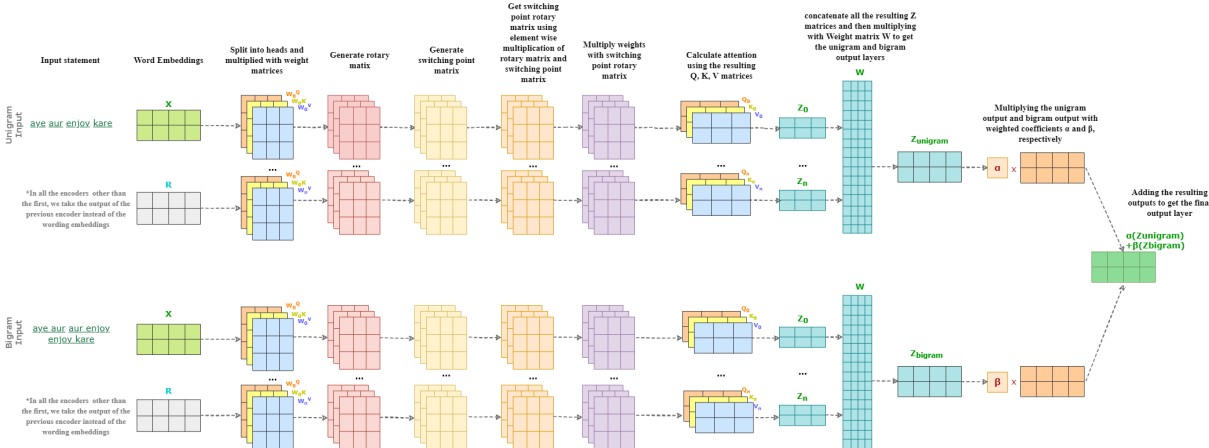

Figure 4: CONFLATOR architecture within the encoder layer. It depicts how the unigrams and bigrams of the input statement are passed as inputs to our encoder decoder architecture. In this framework, we generate a rotary matrix and a switching point matrix. By performing element-wise multiplication of the aforementioned matrices, we get our proposed novel switching point based rotary matrix. We represent the embeddings as complex numbers and their positions as pure rotations that we apply to them with the help of our switching point based rotary matrix. Then, upon getting the output layers for unigram and bigram statements separately. We introduce weighted coefficients $a$ and $b$ for unigram outputs and bigram outputs, respectively. We get our final output layer by adding these weighted unigram and bigram outputs.

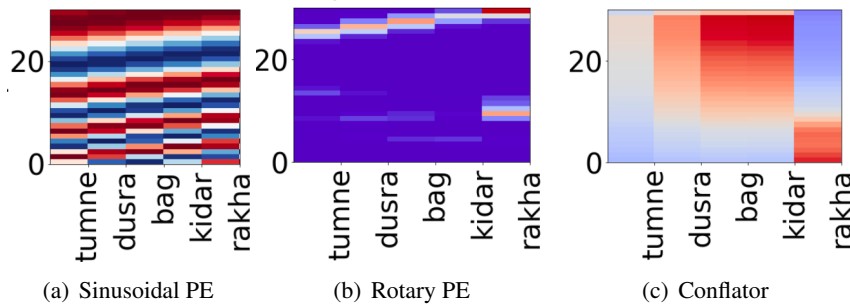

| (a) Sinusoidal PE | (b) Rotary PE | (c) Conflator |

Figure 5: CONFLATOR is able to differentiate words coming from different positions and give high attention when a switching point occurs (at $bag_{EN}$ and $kidar_{HI}$) while the other models cannot do so.

## 6.3 CONFLATOR Architecture

The local dependencies for Unigram and Bigram (Word2Vec trained from scratch) along with unigram and bigram SPRM are fed to a 6-headed Multi-Head attention (MHA) in each encoder layer of the transformer separately, resulting in 2 attention matrices. We introduce 2 learnable parameters $\alpha$ and $\beta$ that are used as weight coefficient for the unigram and bigram matrix respectively. The final matrix is passed to the decoder layer. The embedding and architecture in depicted in figs. 3 and 4.

## 7 Experiments and Results

For our base models, each training step takes about 0.5 seconds. We train the base models for a total of 100,000 steps or 12 hours. For the big models like bigram and SPM-based models, the step time is 1.0 seconds. The big models were trained for 250,000

| CMI Range | Transformer | GPT-2 | BERT | Conflator |
|---|---|---|---|---|
| **0-10** | 1018.54 | 823.71 | 666.48 | 492.96 |
| **11-20** | 1210.11 | 967.01 | 782.19 | 501.44 |
| **21-30** | 1401.37 | 1334.72 | 1007.34 | 544.71 |
| **31-40** | 2688.00 | 2334.73 | 1007.34 | 800.62 |
| **41-50** | 4421.22 | 3905.87 | 4337.02 | 1095.12 |
| Average | 2147.85 | 1873.20 | 1701.49 | **578** |

Table 2: Perplexity comparison between different models based on ranges of CMI. Lower Perplexity is better.

steps (2 days). We use ADAM optimizer with $\beta_1 = 0.9$, $\beta_2 = 0.98$ and $\epsilon = 1e^{-9}$. We use the method of varying the learning rate over the course of training from Vaswani et al. (2017b).

We use two types of regularization during our training process: We apply dropout to the output of each encoder and decoder layer followed by

| Models | Positional representation | | | | | | | Bigram | F1 (%) |
|---|---|---|---|---|---|---|---|---|---|
| | Sin/Cos | Index | Dynamic | SPI | Relative | RM | SPRM | | |
| Word2Vec + LSTM | ✗ | ✗ | ✗ | ✗ | ✗ | ✗ | ✗ | ✗ | 56 |
| BERT | ✓ | ✗ | ✗ | ✗ | ✗ | ✗ | ✗ | ✗ | 60 |
| 3HA + Sinusoidal PE | ✓ | ✓ | ✗ | ✗ | ✗ | ✗ | ✗ | ✗ | 74.34 |
| 3HA + Dynamic PE | ✗ | ✓ | ✓ | ✗ | ✗ | ✗ | ✗ | ✗ | 75.02 |
| 3HA + Relative PE | ✗ | ✗ | ✗ | ✗ | ✓ | ✗ | ✗ | ✗ | 75.32 |
| 3HA + Rotary PE | ✓ | ✓ | ✗ | ✗ | ✗ | ✓ | ✗ | ✗ | 76.04 |
| SOTA (PESTO) | ✗ | ✗ | ✓ | ✓ | ✓ | ✗ | ✗ | ✗ | 75.6 |
| Unigram SP Relative (USPR) | ✗ | ✗ | ✗ | ✓ | ✓ | ✗ | ✗ | ✗ | 75 |
| Bigram SP Relative BSPR) | ✗ | ✗ | ✓ | ✓ | ✓ | ✗ | ✗ | ✓ | 75 |
| Unigram SPRoPE + Good Tuning | ✓ | ✓ | ✗ | ✓ | ✗ | ✓ | ✓ | ✗ | 74.6 |
| Unigram SPRoPE | ✓ | ✓ | ✗ | ✓ | ✗ | ✓ | ✓ | ✗ | 75 |
| Conflator (BSPRoPE) | ✓ | ✓ | ✗ | ✓ | ✗ | ✓ | ✓ | ✓ | 76.23 |
| Conflator with StableLM | ✓ | ✓ | ✗ | ✓ | ✗ | ✓ | ✓ | ✓ | 76.11 |
| Conflator with Alpaca | ✓ | ✓ | ✗ | ✓ | ✗ | ✓ | ✓ | ✓ | 75.69 |
| **Conflator with LLaMA** | ✓ | ✓ | ✗ | ✓ | ✗ | ✓ | ✓ | ✓ | **76.45** |

Table 3: Results of various position sensitive experiments for **Sentiment Analysis** on CM text. *n*HA refers to n-headed attention.

| Models | Positional representation | | | | | | | Bigram | BLEU |
|---|---|---|---|---|---|---|---|---|---|
| | Sin/Cos | Index | Dynamic | SPI | Relative | RM | SPRM | | |
| 3HA + Sinusoidal PE | ✓ | ✓ | ✗ | ✗ | ✗ | ✗ | ✗ | ✗ | 17.2 |
| 3HA + Dynamic PE | ✗ | ✓ | ✓ | ✗ | ✗ | ✗ | ✗ | ✗ | 17.9 |
| 3HA + Relative PE | ✗ | ✗ | ✗ | ✗ | ✓ | ✗ | ✗ | ✗ | 18.4 |
| 3HA + Rotary PE | ✓ | ✓ | ✗ | ✗ | ✗ | ✓ | ✗ | ✗ | 24.9 |
| SOTA (IIITH-mrinaldhar) | ✗ | ✗ | ✓ | ✓ | ✓ | ✗ | ✗ | ✗ | 28.4 |
| Unigram SP Relative (USPR) | ✗ | ✗ | ✗ | ✓ | ✓ | ✗ | ✗ | ✗ | 9.8 |
| Bigram SP Relative (BSPR) | ✗ | ✗ | ✓ | ✓ | ✓ | ✗ | ✗ | ✓ | 7.6 |
| Unigram SPRoPE | ✓ | ✓ | ✗ | ✓ | ✗ | ✓ | ✓ | ✗ | 29.1 |
| Conflator (BSPRoPE) | ✓ | ✓ | ✗ | ✓ | ✗ | ✓ | ✓ | ✓ | 25.16 |
| Conflator with StableLM | ✓ | ✓ | ✗ | ✓ | ✗ | ✓ | ✓ | ✓ | 29.06 |
| Conflator with Alpaca | ✓ | ✓ | ✗ | ✓ | ✗ | ✓ | ✓ | ✓ | 29.89 |
| **Conflator with LLaMA** | ✓ | ✓ | ✗ | ✓ | ✗ | ✓ | ✓ | ✓ | **30.15** |

Table 4: Results of position sensitive experiments for **Machine Translation** on CM text. Higher BLEU is better.

Normalization. In addition, we apply dropout and normalization to the sums of the word embeddings and the positional encodings in both the encoder and decoder layers. We use a rate of $P_{drop} = 0.2$.

**Intrinsic Evaluation:** The perplexity scores of baseline language models in comparison with CONFLATOR on code-mixed language modeling task are shown in 2. We see that our model performs much better than other models.

**Extrinsic Evaluation:** We evaluate our model on two downstream tasks: (i) sentiment analysis, and (ii) machine translation. For sentiment analysis, (Table. 3) we use the data provided by Patwa et al. (2020a). CONFLATOR achieves 76.23% F1 score and outperforms the SOTA (Ali et al., 2022). The main reason for this is learning SP by aggregating with the help of rotary positional encoding with a variable length MHA framework. For the machine translation (Table 4), we use the data provided by (Dhar et al., 2018). We achieve 29.1 bleu score and outperform the SOTA (Dhar et al., 2018) using the Unigram SPRoPE model which is able to learn the patterns of language mixing with the help of

switching point based rotary positional encoding.

## 8 Conclusion & Takeaways

In this work, we report experiments on *Hinglish* sentiment analysis and Machine translation problems through the lens of language modeling. Our contribution could be seen as following:
(i) We introduce the idea of switching point based rotary positional encoding. Whenever a switching point is encountered, we incorporate rotation change to learn the patterns of language mixing.
(ii) We introduce CONFLATOR, a neural language modeling approach for code-mixed languages. CONFLATOR tries to learn better representations by means of switching point-based rotary positional encoding, initially at unigram level and then at bigram level.
(iii) We empirically prove that CONFLATOR is learning the patterns of code-mixing which other models with different positional encodings prove unsuccessful, as shown in Figure 5.
(iv) It is also noteworthy that CONFLATOR achieves comparable to SOTA results even without any pre-trained heavy language model.

## 9 Limitations

Although our bigram model achieves SOTA on sentiment analysis using unigram, it is slightly behind the bigram model when it comes to machine translation, where using bigram at the decoder level resulted in poor performance. Despite conducting extensive experiments, there lacks a detailed explanation on why the bigram-based approach for MT fails. Future experiments will focus on exploring or understanding the issue of bigrams for MT and coming up with a solution for the same.

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
