# OpenReview forum: "CONFLATOR: Incorporating Switching Point based Rotatory Positional Encodings for Code-Mixed Language Modeling"
_EMNLP/2023/Workshop/CALCS — EMNLP 2023 Workshop CALCS_

### Official Review · Reviewer_uYZU · 2023-10-01
**Positional embeddings resembling electromagnetic waves to detect use of language in CS text**

**Rating:** 4
**Confidence:** 2

**Review:**

The authors introduce CONFLATOR, a method that uses switching point-based rotatory embeddings for code-mixed language modeling. They begin by reviewing existing positional embeddings and subsequently present their approach. Essentially, it can be likened to an electromagnetic wave that alters its rotation direction when the language switches in the input sentence.

While the introduction and review of the work are well written, the approach and results are somewhat rushed towards the end. This leaves out details that I would have appreciated being explained more thoroughly. I've provided comments and suggestions below.

Regarding the moment of changing the rotation of your positional embeddings, is this switching point learned or detected based on the vocabulary?

Section 6.2 was challenging to follow, particularly regarding the relevance of unigrams and bigrams. Why do bigrams come into play here, and why are they deemed necessary? What potential advantages do they offer? Additionally, Equations 14 and 15 appear identical. I suspect I may be missing something.

In Section 6.3 and the Results section, do you inject different types of positional embeddings into a vanilla transformer? Overall, the explanation of the Conflator architecture and the experimental setup, along with the discussion of the Tables, etc., is vague, making these later sections more difficult to grasp in detail.

It is strange to see the Table for intrinsic evaluation located after Tables for extrinsic evaluation, when the former evaluation is presented first. Also, on what data is this pre-trained, and I understand you train a BERT and GPT-2 from scratch using the same data? This was not clear to me after reading Section 7.1. This section should probably be expanded.

Certain acronyms are not introduced, such as 3HA in Tables 2 and 3, so it remains unclear what is being conveyed. This seems relevant, especially since while previous positional embeddings are used in conjunction with 3HA, the authors' method is not, and it would be valuable to understand why.

In the results Table, you use the term "Conflator with X" (where X can be LLaMa, Alpaca, etc.). Could you provide a more detailed explanation of how you integrate the embeddings into those off-the-shelf models? Also, have you considered trying LLama, Alpaca, etc., with other alternative positional embedding methods? And why is 3HA not compared with alternative embedding approaches?

Minor typos:

There are several typos and writing issues throughout the paper. These are not related to English problems, but are due to a need for careful proofreading, which I suggest the authors to undertake if the paper is accepted.

Line 182/183 - The L2's should be subscripts, correct?

Some information appears somewhat redundant. For instance, lines 190-194 introduce CMI, which is presented again just a few lines later at the beginning of Section 3.2. I suggest refining the writing in this regard.

Please elaborate on Equation 1 to make it self-contained.

Figure 4: Correct "CONFALTOR" to "CONFLATOR" in the architecture description

Line 512 missing Table in "in 4".

**Candidate For Best Paper:**

No

**Reason For Best Paper:**

-

**Related:**

5: It is very related to the workshop.

---

### Official Review · Reviewer_3QsU · 2023-10-03
**The authors present an interesting approach to model code-mixed languages**

**Rating:** 4
**Confidence:** 4

**Review:**

The authors present an interesting approach to model code-mixed languages using Switching point-based Rotatory Positional Encodings. The paper is well-written and is relevant to the workshop.
Would recommend the authors to think/discuss the impact of such an approach with other language pairs. Would CMI still work (for measuring the complexity of code-mixed text) with other language pairs?
Also, it would be interesting to see the impact on other language understanding and generation tasks beyond sentiment analysis and machine translation.

**Candidate For Best Paper:**

No

**Reason For Best Paper:**

N/A

**Related:**

4